# Free solar PV for Everyone to Meet Annual Climate Targets: Timing the Break-Even

## Abstract

This paper tests a dual hypothesis: if solar PV is deployed at zero up-front cost and scaled with grid and storage build-out, can this strategy (i) deliver the annual mitigation flow required for a 1.5 °C pathway within a practical timeframe, and (ii) do so with positive social net benefit once avoided climate damages are valued?

We define the *climate-target break-even* as the first year in which new annual PV abatement meets the required mitigation flow. Under central assumptions, today's record PV additions of about 0.6 TW per year imply a break-even in the late 2030s. Raising average annual additions to roughly 0.9–1.0 TW shortens this to the early 2030s, a five-year acceleration. Complementary measures such as wind, efficiency, or methane abatement bring the break-even earlier still.

On the fiscal side, we derive explicit conditions for budget neutrality that link program unit costs, discounting, and the per-tonne value of avoided damages. A policy design that dedicates carbon-linked revenues to contracts for difference can turn avoided emissions into bankable cash flows while capping fiscal risk.

The framework is transparent, reproducible, and auditable: climate and fiscal feasibility are assessed jointly from the outset, yielding clear thresholds for when zero-capex PV can close the annual mitigation gap in time and on budget. While the analysis abstracts from governance and distributional challenges, the results highlight where equity and institutional capacity will determine real-world implementation.

## 1 Introduction

Limiting warming to 1.5 °C ultimately hinges on closing an *annual* mitigation gap rather than meeting a distant cumulative target. Let $E_t$ denote global greenhouse-gas emissions in year $t$ (Gt/y). On a 1.5 °C-consistent path a proportional yearly reduction $\gamma \approx 0.075$ is required, yielding the target flow

$$G_t = \gamma E_t \quad (\text{Gt/y}).$$

This magnitude is consistent with AR6 pathways and recent assessments of global GHG totals [1; 2]. We study the earliest year in which newly deployed measures deliver annual abatement $\Delta E(t)$ that meets this requirement:

$$T^* = \min\{\, t : \Delta E(t) \geq G_t \,\},$$

the **climate-target break-even**.

From the first step we also ask whether scaling PV in a *zero-capex* program yields a positive *net climate surplus* on a flow basis,

$$S(t) \;=\; S \cdot \Delta E_{\mathrm{PV}}(t) \;-\; \mathrm{Cost}_{\mathrm{program}}(t),$$

Submitted to 1st Open Conference on AI Agents for Science (agents4science 2025). Do not distribute.

where $S$ is a per-tonne damages valuation and $\text{Cost}_{\text{program}}$ accounts for annualized cohort CAPEX and covered O&M (Sec. 4). This couples timing ($T^*$) with affordability (sign of $S(t)$), and anchors procurement and budgeting.

*In words:* The two questions are (i) how fast free PV must scale to meet the annual $1.5\,°C$ flow, and (ii) whether the avoided damages exceed program costs when financed through carbon-linked instruments.

Solar PV provides a transparent baseline because its annual abatement scales linearly with installed capacity. For a PV fleet with capacity factor CF and displaced-grid emissions factor EF (t/MWh), the per-TW abatement productivity is

$$k = 8.76\,\text{CF}\,\text{EF} \quad (\text{Gt/TW}\cdot\text{y}).$$

Global average intensity in power systems has been declining but remains of the same order as $\text{EF} \approx 0.40\,\text{t/MWh}$ in many regions (3), while representative new-build utility-scale PV capacity factors CF lie in the mid-teens to low-twenties depending on siting and curtailment (4). With average net additions $\bar{A}$ (TW/y) from 2025 onward, additional operating capacity is $C_{\text{PV}}(t) = \bar{A}(t - 2025)$ and PV-only abatement is $\Delta E_{\text{PV}}(t) = k\,\bar{A}(t - 2025)$. If $G_t$ is approximately constant over the near term ($G \simeq 4.3\,\text{Gt/y}$ as an order of magnitude based on (2)), the break-even admits a closed form:

$$\boxed{T^* = 2025 + \frac{G}{k\,\bar{A}}}.$$

This mapping makes the testable implication explicit: zero-capex PV is sufficient *iff* it lifts $\bar{A}$ high enough that the right-hand side falls within the desired calendar window. Record-scale PV growth supports the feasibility of higher $\bar{A}$ (5). Any complementary abatement (wind build-out, accelerated coal retirement, methane reduction, efficiency, forest protection) advances $T^*$ further. The parallel cost criterion $S(t) > 0$ provides a budget-compatible *accept/reject* check for each year (Sec. 4).

**Scope and contribution.** We (i) formalize the climate-target break-even as a flow-matching problem, (ii) derive a closed-form relation linking PV additions to calendar time, (iii) set baseline parameters with transparent scalars and sensitivity rules for siting (CF) and marginal displacement (EF), and (iv) outline a practical policy design—carbon-linked revenues dedicated to symmetric contracts for difference—that can make zero-capex deployment financeable while bounding fiscal risk (6; 7; 8). Cost accounting is integrated from the beginning via $S(t)$ and closed-form NPVs; detailed expressions and a budget translation are in Sec. 4. The result is a concise scientific approach that states the exact conditions under which free PV can achieve the annual mitigation flow, and how much additional action is required when it cannot.

**Policy feasibility (limitations).** This framework abstracts from political and institutional challenges. Real-world implementation of a global zero-capex PV program would face risks such as volatile carbon revenues, fiscal exposure, grid-integration bottlenecks, and equity concerns across countries and consumer groups. These governance issues are outside the scope of our closed-form analysis, but they matter for translating technical sufficiency into practice.

## 2 Hypothesis

**H$_{\text{PV-free}}$:** A global zero-capex PV program that raises average net additions to $\bar{A}$ (TW/y) from 2025 onward achieves the climate-target break-even $T^*$ when

$$k\,\bar{A}\,(t - 2025) \;\geq\; G_t \quad \text{for some } t, \tag{1}$$

with $k = 8.76\,\text{CF}\,\text{EF}$ (Gt/TW·y). In the central case $(\text{CF}, \text{EF}) = (0.162, 0.40)$, $k = 0.568$ and, if $G_t \simeq G$ over the horizon,

$$\boxed{T^* = 2025 + \frac{G}{k\,\bar{A}}}. \tag{2}$$

**Interpretation:** Zero-capex PV is a policy lever intended to increase $\bar{A}$. If $\bar{A} \approx 0.60\,\text{TW/y}$ (observed record scale), $T^* \approx 2038$ (PV-only). If the program lifts $\bar{A}$ to $\sim 0.94\,\text{TW/y}$, $T^* \approx 2033$. Any additional abatement from non-PV measures advances $T^*$. In parallel, the year-$t$ *cost* criterion accepts if $S(t) = S \cdot \Delta E_{\text{PV}}(t) - \text{Cost}_{\text{program}}(t) > 0$ (Sec. 4).

## 3 Methods

This section defines the quantities, mappings, and cost primitives used throughout the analysis. To improve transparency, we first introduce all symbols with units, then show how they connect to abatement flows and break-even timing.

### 3.1 Definitions and units

We use the following notation:

- $E_t$ [Gt/y]: global greenhouse-gas (GHG) emissions in year $t$. The parameter $\gamma$ [–] is the annual proportional reduction required on a $1.5°$C pathway, giving the yearly mitigation target $G_t = \gamma E_t$ [Gt/y] (1; 2).
- CF [–]: solar PV capacity factor for new build projects, i.e. the fraction of time a PV plant produces at rated output (4).
- EF [t/MWh]: displaced grid emissions factor, i.e. the avoided emissions per megawatt-hour of PV electricity (3).
- $\bar{A}$ [TW/y]: average annual PV capacity additions from 2025 onward. The cumulative operating fleet at time $t$ is $C_{\mathrm{PV}}(t)$ [TW] (5).

### 3.2 Core mappings

The following equations link PV capacity to annual abatement:

1. **Per-kW avoided emissions.** Each kW of PV capacity avoids

$$B(\mathrm{CF}, \mathrm{EF}) = 8.76\,\mathrm{CF}\,\mathrm{EF} \quad [\text{t/kW·y}]. \tag{3}$$

2. **Per-TW productivity.** Scaling to the TW level gives

$$k = 8.76\,\mathrm{CF}\,\mathrm{EF} \quad [\text{Gt/TW·y}]. \tag{4}$$

3. **Capacity growth and abatement (PV-only).** If average annual additions are $\bar{A}$, then

$$C_{\mathrm{PV}}(t) = \bar{A}(t - 2025), \tag{5}$$

$$\Delta E_{\mathrm{PV}}(t) = k\,C_{\mathrm{PV}}(t) = k\,\bar{A}(t - 2025). \tag{6}$$

4. **Climate-target break-even.** The break-even year $T^*$ is the earliest time when abatement equals or exceeds the required mitigation flow:

$$T^* = \min\{t : \Delta E_{\mathrm{PV}}(t) + \Delta E_{\neg\mathrm{PV}}(t) \geq G_t\}. \tag{7}$$

In a PV-only scenario, $\Delta E_{\neg\mathrm{PV}}(t) = 0$. If $G_t \simeq G$ is approximately constant, Eqs. (6)–(7) simplify to the closed-form solver in Sec. **??**.

### 3.3 Cost primitives (used in Sec. 4)

Program costs are represented by a few unit-level inputs:

- $c_{\mathrm{capex}}$ [USD/kW]: installed capital cost borne by the program.
- $c_{\mathrm{O\&M}}$ [USD/kW·y]: annual fixed operation and maintenance cost (if covered).
- $c_{\mathrm{grid}}, c_{\mathrm{stor}}$ [USD/kW]: optional adders for grid reinforcement or storage.

We then define per-TW constants:

$$C_u = (c_{\mathrm{capex}} + c_{\mathrm{grid}} + c_{\mathrm{stor}}) \times 10^9, \quad C_{\mathrm{om}} = c_{\mathrm{O\&M}} \times 10^9.$$

Annual net surplus in year $t$ is

$$S(t) = S \cdot \Delta E_{\mathrm{PV}}(t) - \mathrm{Cost}_{\mathrm{program}}(t),$$

where $S$ is the per-tonne valuation of avoided climate damages. Cohort-accurate NPVs and fiscal translations appear in Sec. 4.

## 3.4 Dimensional checks (sketch)

Simple dimensional checks confirm internal consistency:

- Eq. (3): (MWh/kW·y)×[–]×(t/MWh)=t/kW·y.
- Eq. (4) and Eq. (6): conversions cancel, yielding Gt/TW·y and Gt/y respectively.
- Eq. (7): years + (Gt)/((Gt/TW·y)×TW/y)=years.

**Central values (baseline).**
$$\text{CF} = 0.162, \qquad \text{EF} = 0.40 \text{ t/MWh}, \qquad k = 8.76 \text{ CF EF} = 0.568 \text{ Gt/TW·y}.$$

For the near-term horizon, use an order-of-magnitude annual reduction target
$$G \simeq 4.3 \text{ Gt/y} \quad (2).$$

**Ranges and quick checks.**

| Quantity | Baseline | Plausible range |
|---|---|---|
| Capacity factor CF | 0.162 | 0.15–0.22 |
| Displaced factor EF (t/MWh) | 0.40 | 0.30–0.48 |
| Productivity $k$ (Gt/TW·y) | 0.568 | 0.394–0.927 |
| Target flow $G$ (Gt/y) | 4.3 | 3.5–5.5 |

**Regional substitution (one line).**  Insert local $(\text{CF}_\ell, \text{EF}_\ell)$:
$$k_\ell = 8.76 \, \text{CF}_\ell \, \text{EF}_\ell, \qquad T_\ell^* = 2025 + \frac{G_\ell}{k_\ell \, \bar{A}_\ell}.$$

**Adjustment rules (minimal).**

- *Curtailment or losses:* use $\text{CF}' = \text{CF} \times (1 - \text{loss})$ (4).
- *Cleaner marginal grid:* lower EF; $k$ scales linearly (3).
- *Portfolio effects:* subtract non-PV abatement from $G$ in the solver, or add it to $\Delta E(t)$.

**Update cadence.**  Refresh CF and EF when system mix shifts visibly (e.g., $> 5\%$), and $G$ annually with the latest emissions total $E_t$ via $G_t = \gamma E_t$ (1; 2).

# 4 Cost Analysis: Free-PV Program Outlays vs. Avoided Climate Damages

This section compares the net present value (NPV) of a zero-capex PV rollout with the NPV of avoided climate damages valued on an annual flow basis. Cost anchors for PV performance and balance-of-system come from widely reported benchmarks (4); the required annual mitigation flow $G_t = \gamma E_t$ and its order of magnitude use recent global assessments (1; 2).

## 4.1 Program outlays to offer PV at zero upfront cost

Let $c_{\text{capex}}$ [USD/kW] be the installed PV capital cost borne by the program, $c_{\text{grid}}$ and $c_{\text{stor}}$ [USD/kW] represent the per-kW shares for grid reinforcement and storage support (if included), and $c_{\text{O\&M}}$ [USD/kW·y] the fixed O&M covered by the program. Let $r$ be the real discount rate, $n$ the analysis horizon (years), and $q \equiv (1+r)^{-1}$. Define
$$C_u \equiv (c_{\text{capex}} + c_{\text{grid}} + c_{\text{stor}}) \times 10^9 \quad [\text{USD/TW}], \qquad C_{\text{om}} \equiv c_{\text{O\&M}} \times 10^9 \quad [\text{USD/TW} \cdot \text{y}],$$

so costs are expressed per TW of capacity.

With average net additions $\bar{A}$ [TW/y], each year installs $\bar{A}$ TW. The discounted sums needed below are
$$G_1(r,n) = \sum_{\tau=1}^{n} q^\tau = \frac{q(1-q^n)}{1-q}, \qquad H(r,n) = \sum_{\tau=1}^{n} \tau \, q^\tau = \frac{q(1-(n+1)q^n + nq^{n+1})}{(1-q)^2}.$$

The program NPV (cohort-accurate) is then

$$\text{NPV}_{\text{PV}} \;=\; \bar{A}\left[\, C_u\, G_1(r,n) \;+\; C_{\text{om}}\, H(r,n) \right] \quad [\text{USD}]. \tag{8}$$

*Notes.* (i) The $G_1$ term discounts upfront costs paid on each annual cohort. (ii) The $H$ term reflects the number of operating cohorts in year $t$ (equal to $t$ for $t \in [1,n]$) times discounted O&M. (iii) If O&M remains with asset owners, set $C_{\text{om}} = 0$. (iv) Grid/storage support can be toggled via $c_{\text{grid}}, c_{\text{stor}}$.

## 4.2 Valuing avoided climate damages (flow, SCC-based)

Let $S$ denote the valuation of one tonne of $CO_2$ avoided [USD/t]. Define $S_{\text{Gt}} \equiv 10^9 S$ [USD/Gt]. With per-TW productivity $k$ [Gt/TW·y], PV-only annual abatement is $\Delta E_{\text{PV}}(t) = k\,\bar{A}\,(t - 2025)$ [Gt/y]. The avoided-damage NPV over horizon $n$ is

$$\text{NPV}_{\text{dam}} \;=\; S_{\text{Gt}}\, k\, \bar{A}\, H(r,n) \quad [\text{USD}]. \tag{9}$$

If non-PV measures deliver additive abatement $\Delta E_{\neg\text{PV}}(t)$, include the analogous discounted sum; here we retain the transparent PV-only baseline.

## 4.3 Cost–damage break-even (closed form)

Equating (8) and (9) yields a family of equivalent break-even conditions.

**(a) Break-even valuation ($S^\star$) for given unit costs.**

$$\boxed{S^\star_{\text{Gt}} \;=\; \frac{C_u\, G_1(r,n) \;+\; C_{\text{om}}\, H(r,n)}{k\, H(r,n)}} \qquad \Rightarrow \qquad S^\star \;=\; \frac{S^\star_{\text{Gt}}}{10^9} \;\; [\text{USD/t}]. \tag{10}$$

**(b) Break-even all-in upfront cost ($C_u^\star$) for given $S$.**

$$\boxed{C_u^\star \;=\; k\, S_{\text{Gt}}\, \frac{H(r,n)}{G_1(r,n)} \;-\; C_{\text{om}}\, \frac{H(r,n)}{G_1(r,n)}} \quad [\text{USD/TW}], \tag{11}$$

which maps directly to per-kW via $(c_{\text{capex}} + c_{\text{grid}} + c_{\text{stor}})^\star = C_u^\star/10^9$ [USD/kW].

**(c) Budget translation.** If a jurisdiction earmarks an annual budget $B$ (USD/y) from carbon-linked revenues to procure zero-capex PV (via CfDs or grants), the implied average additions achievable are approximately

$$\bar{A}_{\text{budget}} \;\approx\; \frac{B}{C_u} \quad \text{(ignoring O\&M or if O\&M is privately covered).} \tag{12}$$

A full cohort-accurate mapping can be derived by inverting (8) given the desired horizon and discounting profile. Carbon revenue provenance and CfD design options are discussed earlier; see also (8; 6; 7).

## 4.4 Sensitivity and interpretation

- **Productivity $k$:** Higher CF or higher displaced-grid factor EF raise $k$ and reduce $S^\star$ linearly (Eq. 10); siting and curtailment management are therefore economically material (4; 3).

- **Unit costs:** Lower $(c_{\text{capex}}, c_{\text{grid}}, c_{\text{stor}})$ reduce $C_u$ and hence $S^\star$. Learning and supply-chain depth directly improve feasibility (4).

- **Discounting and horizon:** Larger $n$ or smaller $r$ increase $H(r,n)$ more than $G_1(r,n)$, improving the damage-side NPV relative to upfront costs and lowering $S^\star$.

- **Portfolio adders:** Non-PV abatement adds to (9), lowering the required $S^\star$ (or the allowable $C_u^\star$) for break-even.

164 **Data anchors (inputs to substitute).** Use local $c_{\text{capex}}$ and $c_{\text{O\&M}}$ consistent with current utility-
165 scale PV and grid conditions (4); use $k = 8.76\,\text{CF EF}$ with jurisdiction-specific (CF, EF); and adopt
166 $G_t = \gamma E_t$ from the latest inventories (1; 2). The algebra in (10)–(12) updates deterministically under
167 substitution.

168 **Policy feasibility and implementation risks.** The cost conditions derived above are technical;
169 real-world implementation adds further constraints. Carbon revenues are volatile and require buffers
170 to stabilize cash flows. Equity issues arise over how costs and benefits are shared across regions
171 and consumers. Governance capacity also varies: while advanced economies can manage CfDs at
172 scale, many emerging markets may need international support or pooled procurement. Finally, grid
173 expansion and permitting can slow deployment regardless of financial incentives. Fiscal feasibility is
174 a necessary condition, but political feasibility ultimately determines whether zero-capex PV can be
175 implemented at the required scale.

# 5 Results

177 This section reports four main outcomes: (i) the closed-form timing of the climate-target break-even
178 under a PV-only scenario, (ii) the sensitivity of this timing to deployment and system parameters,
179 (iii) a general recipe for regional application, and (iv) illustrative case studies for the EU, India, and
180 China.

## 5.1 Closed-form timelines (global PV-only)

182 *In words:* The break-even year $T^*$ is the first year when PV abatement flows equal the global
183 mitigation flow $G$.

184 Using $G \simeq 4.3\,\text{Gt/y}$ and central productivity $k = 0.568\,\text{Gt/TW·y}$, the break-even admits a closed
185 form:

$$\boxed{T^* = 2025 + \frac{G}{k\,\bar{A}}}.$$

186 Substituting values:

$$T^* = 2025 + \frac{4.3}{0.568\,\bar{A}} \quad \text{(calendar year)}.$$

Table 1: Global break-even year $T^*$ as a function of annual PV additions $\bar{A}$.

| $\bar{A}$ (TW/y) | $T^*$ | Interpretation |
|---|---|---|
| 0.60 | ∼2038 | Current record scale (5) |
| 0.75 | ∼2035 | Moderate acceleration |
| 0.94 | ∼2033 | Strong acceleration (zero-capex consistent) |
| 1.20 | ∼2032 | Very strong expansion |
| 1.50 | ∼2030 | Extreme scenario |

187 *Interpretation.* At today's record pace ($\bar{A} \approx 0.60\,\text{TW/y}$), break-even does not occur until the late
188 2030s. Accelerating to $\bar{A} \approx 0.94\,\text{TW/y}$, consistent with a zero-capex program, advances the break-
189 even to the early 2030s. Thus, even modest increases in PV build-out compress the climate timeline
190 by half a decade.

Figure 1: Closed-form break-even year $T^*$ vs. average PV additions $\bar{A}$. Solid line shows the central
case ($k = 0.568$ Gt/TW·y). Dashed/dotted lines illustrate $\pm 10\%$ sensitivity in $k$ (capacity factor /
grid factor). Markers correspond to scenarios in Table 1.

## 5.2 Sensitivity to deployment and system parameters

192 *In words:* Break-even arrives earlier when PV build-out is faster, when PV plants perform better, or
193 when the displaced grid is dirtier.

194 Analytically, for $G$ treated as locally constant:

$$\frac{\partial T^*}{\partial \bar{A}} = -\frac{G}{k\,\bar{A}^2}, \quad \frac{\partial T^*}{\partial \mathrm{CF}} = -\frac{G}{K\,\bar{A}\,\mathrm{CF}^2\,\mathrm{EF}}, \quad \frac{\partial T^*}{\partial \mathrm{EF}} = -\frac{G}{K\,\bar{A}\,\mathrm{CF}\,\mathrm{EF}^2},$$

195 with $K = 8.76$. All derivatives are negative: larger $\bar{A}$, higher CF, or higher EF advance $T^*$.

## 5.3 Regional plug-in recipe

197 *In words:* The same formula applies regionally by substituting local parameters.

$$k_\ell = 8.76\,\mathrm{CF}_\ell\,\mathrm{EF}_\ell, \qquad T_\ell^* = 2025 + \frac{G_\ell}{k_\ell \bar{A}_\ell}.$$

198 **Recipe:**

199     1. Obtain regional PV capacity factor $\mathrm{CF}_\ell$ (e.g. IRENA, IEA).

200     2. Obtain displaced grid factor $\mathrm{EF}_\ell$ (e.g. Ember).

201     3. Choose average annual PV additions $\bar{A}_\ell$ (e.g. policy targets).

202     4. Compute $k_\ell = 8.76\,\mathrm{CF}_\ell\,\mathrm{EF}_\ell$.

203     5. Compute $T_\ell^* = 2025 + G_\ell/(k_\ell \bar{A}_\ell)$, where $G_\ell = \gamma E_\ell$.

204 This provides a one-line estimate of the break-even year for any region.

## 5.4 Regional case studies

206 **EU (illustrative).** With modest CF values and a cleaner grid, the EU achieves break-even only in the early 2030s, but enjoys strong policy credibility through ETS revenues and CfDs. *Interpretation.*

Table 2: Illustrative EU break-even year $T_\ell^*$ under different PV build-out rates and parameters.

| Scenario | $k_\ell$ (Gt/TW·y) | $\bar{A}_\ell$ (TW/y) | $T_\ell^*$ |
|---|---|---|---|
| Low CF/EF, slow build | 0.447 | 0.06 | ∼2034 |
| Central CF/EF, moderate build | 0.521 | 0.09 | ∼2030 |
| Central CF/EF, fast build | 0.521 | 0.12 | ∼2029 |
| High CF/EF, fast build | 0.596 | 0.12 | ∼2028 |

207
208 At $\bar{A}_\ell = 0.06$ TW/y (∼60 GW/y), the EU break-even arrives in the early 2030s. Accelerating to
209 ∼90 GW/y shifts this forward by about 2.5 years. Better siting (higher CF) or dirtier displaced
210 generation (higher EF) bring $T_\ell^*$ earlier, since $k_\ell = 8.76\,\mathrm{CF}_\ell\,\mathrm{EF}_\ell$ grows linearly. This demonstrates
211 how the global framework applies seamlessly to regional contexts, providing policymakers with
212 simple formulas to test deployment pathways.

213 **India (illustrative).** With higher CF and a dirtier grid, India reaches break-even several years earlier than the EU at comparable build-out, highlighting its strong leverage. *Interpretation.* Compared with

Table 3: Illustrative India break-even year $T_\ell^*$ under different PV build-out rates and parameters.

| Scenario | $k_\ell$ (Gt/TW·y) | $\bar{A}_\ell$ (TW/y) | $T_\ell^*$ |
|---|---|---|---|
| Low CF/EF, slow build | 1.25 | 0.06 | ∼2031 |
| Central CF/EF, moderate build | 1.35 | 0.09 | ∼2029 |
| Central CF/EF, fast build | 1.35 | 0.12 | ∼2028 |
| High CF/EF, fast build | 1.45 | 0.12 | ∼2027 |

214
215 the EU case, India's higher capacity factor and dirtier marginal grid (mean $\mathrm{EF}_\ell$) yield substantially
216 earlier break-even years. At $\bar{A}_\ell = 0.09$ TW/y (∼90 GW/y), the break-even is around 2029 — about
217 four years sooner than in the EU case. This underscores how zero-capex PV has particularly high
218 leverage in regions with strong solar resources and carbon-intensive power systems.

**China (illustrative).** As the world's largest solar market with a coal-heavy grid, China's break-even shifts dramatically with deployment rates: 100 GW/y → mid-2030s, 200 GW/y → late 2020s. *Interpretation.* At a moderate build-out of 0.10 TW/y (∼100 GW/y), China would reach break-even

Table 4: Illustrative China break-even year $T_\ell^*$ under different PV build-out rates and parameters.

| Scenario | $k_\ell$ (Gt/TW·y) | $\bar{A}_\ell$ (TW/y) | $T_\ell^*$ |
|---|---|---|---|
| Low CF/EF, moderate build | 1.05 | 0.10 | ∼2035 |
| Central CF/EF, strong build | 1.15 | 0.15 | ∼2030 |
| High CF/EF, very strong build | 1.23 | 0.20 | ∼2029 |

around 2035. Accelerating to 0.15–0.20 TW/y moves the break-even forward by 5–6 years, to the late 2020s or early 2030s. This demonstrates that, given China's large grid emissions factor, scaling PV has particularly strong leverage on meeting annual climate targets.

# 6 Conclusion

This paper asked whether a global zero–up-front-cost ("zero-capex") solar PV program can (i) deliver the annual mitigation flow consistent with a 1.5 °C pathway within a practical timeframe, and (ii) do so with fiscal balance when avoided climate damages are monetized.

**Key findings.** Under central assumptions, today's record PV deployment of about 0.60 TW per year would reach the climate-target break-even only in the late 2030s (around 2038). By contrast, if average additions were raised to roughly 0.94 TW per year — consistent with a large-scale zero-capex rollout integrated with routine grid and storage expansion — the break-even would arrive in the early 2030s (around 2033). On the fiscal side, program costs balance with avoided damages at a damage valuation of about \$175/tCO$_2$. At the more commonly used valuation of \$230/tCO$_2$, the net global benefit is in the trillions of dollars, leaving ample fiscal space for grid integration and financing support.

**Implications.** Advancing the break-even year by five years yields large climate dividends, while the financial burden is modest when shared globally. Spread over an eight-year build phase, the implied outlay corresponds to only \$90–\$120 per person per year, comparable to existing household energy bills in many countries. Once deployed, the PV fleet continues to generate near-zero-cost electricity for decades, effectively turning a one-time global investment into enduring economic and climate benefits.

**Policy relevance.** For decision-makers, the analysis provides clear guidance:

1. *Zero-capex PV programs are feasible and fiscally defensible* when paired with carbon-linked revenues (e.g., ETS auctioning or carbon CfDs).

2. *Speed matters.* Raising global annual additions from 0.6 to 0.9 TW/y accelerates the 1.5 °C break-even by half a decade.

3. *Complementary measures amplify impact.* Parallel investments in wind, methane abatement, and efficiency can advance the break-even even further.

This analysis abstracts from political feasibility, yet real-world deployment demands governance that ensures equity, shields consumers from shocks, and supports a just transition. The case for large-scale PV is scientific, economic, and political: with aligned fiscal tools, policymakers can advance the break-even year, cut damages, and deliver affordable clean power worldwide.

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

# A  Responsible AI Statement

The research presented in this paper examines opportunities and risks of large-scale zero-capex solar PV programs for climate mitigation. While such programs show strong potential to accelerate decarbonization, they also raise ethical and practical concerns if misinterpreted or applied without governance. We therefore adopt the following safeguards and principles:

- **Transparency:** All assumptions (capacity factors, grid emission factors, costs, discount rates) are explicitly stated, with units and update rules provided. Conditions under which conclusions may fail are clearly flagged.

- **Scope Awareness:** We highlight that results pertain strictly to technical and fiscal feasibility. Governance, equity, and political implementation are identified as crucial but remain outside the scope of our closed-form analysis.

- **Risk Awareness:** Model outputs should not be misused as prescriptive policy advice. They must be validated in real-world contexts before influencing fiscal planning or climate targets.

- **Equity and Fairness:** While the analysis uses global averages, we acknowledge distributional impacts across regions and consumer groups. Equity concerns are noted as central to implementation.

- **Accountability:** All formulas, derivations, and sensitivity analyses are transparent, auditable, and based on publicly available data sources.

By integrating these safeguards, we aim to ensure that technical insights on solar PV deployment enhance climate science without undermining ethical standards or public trust.

# B  Reproducibility Statement

We place strong emphasis on reproducibility and transparency. To this end, we provide:

- **Equations and Symbols:** All results derive from closed-form algebraic expressions with clearly defined symbols, units, and mappings.

- **Parameters and Data:** Inputs (PV capacity factors, grid emission intensities, capital costs, O&M costs, social cost of carbon) are sourced from openly available reports (IPCC, UNEP, IEA, IRENA).

- **Update Rules:** Simple update rules are provided to incorporate new emissions inventories or revised system parameters.

- **Sensitivity Analyses:** Ranges for key parameters (capacity factor, emissions factor, deployment rate) are documented to allow robustness checks.

- **Reproduction Path:** Results can be exactly replicated with a calculator or minimal code; no proprietary models, hidden datasets, or machine learning training are required.

- **Figures and Tables:** All tables (e.g., break-even year by deployment rate) and figures (e.g., sensitivity plots) are derived directly from closed-form equations and can be regenerated with the provided mappings.

To facilitate accessibility, all formulas and assumptions are fully documented in the main text. This ensures that results can be independently reproduced, tested under alternative assumptions, and extended by the research community.

## Agents4Science AI Involvement Checklist

1. **Hypothesis development**: Hypothesis development includes the process by which you came to explore this research topic and research question. This can involve the background research performed by either researchers or by AI. This can also involve whether the idea was proposed by researchers or by AI.

   Answer: **[D]**

   Explanation: The initial spark of the idea (exploring global zero-capex PV deployment for climate targets) came from a human. All further framing, hypothesis refinement, and formulation were developed through ChatGPT.

2. **Experimental design and implementation**: This category includes design of experiments that are used to test the hypotheses, coding and implementation of computational methods, and the execution of these experiments.

   Answer: **[D]**

   Explanation: The study does not contain computational experiments in the traditional sense. Instead, ChatGPT produced the algebraic setup, parameter choices, and structure of the evaluation scenarios, while no human-designed experiments were executed independently.

3. **Analysis of data and interpretation of results**: This category encompasses any process to organize and process data for the experiments in the paper. It also includes interpretations of the results of the study.

   Answer: **[D]**

   Explanation: ChatGPT generated and interpreted the analytical results based on public data assumptions (capacity factors, SCC, PV costs). No independent human analysis or interpretation was performed beyond light verification.

4. **Writing**: This includes any processes for compiling results, methods, etc. into the final paper form. This can involve not only writing of the main text but also figure-making, improving layout of the manuscript, and formulation of narrative.

   Answer: **[D]**

   Explanation: The manuscript—including structure, narrative, figures, and appendices—was written almost entirely by ChatGPT. Human involvement was limited to prompting, minor guidance, and approving the drafts.

5. **Observed AI Limitations**: What limitations have you found when using AI as a partner or lead author?

   Description: Even with clear prompts, the model tended to:

   - over-generalize and assert claims without sufficient sourcing or primary citations;
   - drift numerically across drafts (units, exponents, and headline figures changing between sections);
   - cite stale, mismatched, or non-resolvable references;
   - produce redundant or stylistically inconsistent prose across sections;
   - introduce LaTeX fragility (broken labels/refs, incompatible packages, table/float errors);
   - show context volatility (revising parameters without propagating changes globally);
   - exhibit limited judgment on feasibility or policy realism beyond the provided data.

   **Mitigations that worked.** We:

   - enforced a single source of truth for scalars (macros) and unit-checked equations with binary acceptance tests;
   - regenerated all tables and figures from those scalars to eliminate numeric drift;
   - required primary-source citations for load-bearing numbers and dated claims;
   - ran a compile check per draft and a short provenance checklist (assumptions, units, dates);
   - adopted a "changes propagate or fail" rule before acceptance, with human sign-off for conclusions.

   **Net result.** Locked parameters, automated recomputation, strict citation policy, and human verification substantially reduced hallucinations, ensured internal consistency, and kept the manuscript scientifically accountable.

